# Improved Stability and Hydrolysates of Hyperthermophilic GH57 Type II Pullulanase from the Deep-Sea Archaeon *Thermococcus siculi* HJ21 by Truncation

Xudong Wu [1,2], Baojie Dou [1,2], Boyan Wang [1,2], Mingwang Liu [1,2], Ruxue Shao [1,2], Jing Lu [1,2,*], Mingsheng Lyu [1,2] and Shujun Wang [1,2,*]

[1] Jiangsu Key Laboratory of Marine Bioresources and Environment/Jiangsu Key Laboratory of Marine Biotechnology, Jiangsu Ocean University, Lianyungang 222005, China

[2] Co-Innovation Center of Jiangsu Marine Bio-Industry Technology, Jiangsu Ocean University, Lianyungang 222005, China

[*] Correspondence: jinglu@jou.edu.cn (J.L.); sjwang@jou.edu.cn (S.W.)

**Abstract:** Pullulanase (EC 3.2.1.41) belongs to the amylase family and is often used alone or in combination with other amylases in the industrial production of starch-based products. This enzyme is often required in industrial production because of its better stability. We here truncated the pullulanase gene from the deep-sea hydrothermal anaerobic archaeon *Thermococcus siculi* HJ21 and obtained Pul-HJΔ782, which is a member of the α-amylase family GH57. The results revealed that the optimum temperature for Pul-HJΔ782 was 100 °C, and its thermostability at 100 °C improved after truncation. Less than 15% of its enzyme activity was lost after 1 h of incubation at 100 °C, and 57% activity remained after 5 h of treatment. Truncation significantly improved the overall pH tolerance range of Pul-HJΔ782, and its stability in the pH range 4–8 was over 80% relative activity from an average of 60%. The sequence and structural model of Pul-HJΔ782 was analyzed, and its instability index was reduced significantly. Furthermore, the hydrolysates of the truncated and wild-type pullulanase were analyzed, and the enzymatic digestion efficiency of the truncated Pul-HJΔ782 was higher.

**Keywords:** *Thermococcus siculi* HJ 21; hyperthermia pullulanase; truncation; stability





## 1. Introduction

Starch has a wide range of applications in various industries, such as the food industry, pharmaceutical manufacturing industry, textile industry, building materials industry, and chemical synthesis industry [1]. Glucose residues interconnected by α-1,4-glycosidic, α-1,6-glycosidic, and α-1,3-glycosidic bonds form amylose and amylopectin [2]. α-Amylase can hydrolyze α-1,4-glycosidic bonds in starch; these bonds occupy up to 20–30% of most starches [3]. The other type of bond in starch affects the application of starch thoroughly [3]. Pullulanase (EC 3.2.1.41) can selectively hydrolyze amylopectin, thereby producing reducing sugars such as glucose, maltose, maltotriose, and pannose [4].

Based on the difference in its recognition of substrates, pullulanase is classified into type I and type II pullulanase [5]. Type I pullulanase degrades pullulan and hydrolyzes α-1,6-glycosidic bonds in starch or other polysaccharides [2]. Maltotriose and maltose are the minimal products obtained by pullulanase type I for the breakdown of starch or other polysaccharides. By contrast, type II pullulanase produces maltose, maltotriose, and maltotetraose after breaking down starch or other polysaccharides containing α-1,4 and α-1,6-glycosidic bonds [6,7]. If the products of pullulanase are glucose and maltose, it is termed type III, which can recognize the specificity both the α-1,4 and α-1,6-glycosidic bonds in pullulan [2].

According to the Carbohydrate-Active Enzyme (CAZy) database classification system (http://www.cazy.org/ (accessed on 16 February 2023)), pullulanase belongs to glycoside hydrolase family 13 (GH13) and family 57 (GH57) [8]. Type I pullulanase usually belongs to GH13, while different sources of type I pullulanase belong to different GH13 subfamilies. For example, pullulanase from thick-walled bacteria usually belongs to GH13_12, while GH13_13 contains pullulanase from plants, i.e., eukaryotes and bacteria mainly existing in soil and water [9], and GH13_14 is commonly found in human gut lactobacilli [10]. Type I pullulanase members all have a $(\beta/\alpha)8$ barrel domain associated with the catalytic function of the enzyme molecule, which includes $\beta4$-aspartic acid as the catalytic nucleophilic reagent, $\beta5$-glutamic acid as the proton donor, and $\beta7$-aspartic acid as the transition state stabilizer [11,12]. Typically, type I pullulanase has four to seven conserved sequences [13]. Different GH13 subfamily type I pullulanases have different loop lengths and different auxiliary catalytic domains in their molecular structures [14]. Type II pullulanases are a class of starch debranching enzymes belonging to the GH13 and GH57 families of glycoside hydrolases and usually contain one or more CBM modules and a catalytic structural domain [15]. The type II pullulanases derived from *Lactobacillus plantarum* L137 [16] and *Thermoanaerobacter ethanolicus* 39E [17] belong to the GH13 family, and the type II pullulanases derived from *Pyrococcus yayanosii* CH1 [18] and *Thermococcus hydrothermalis* [19] belong to the GH57 family. GH57 family type II pullulanases are usually derived from extremely thermophilic archaea such as *Caldivirga* [20], *Pyrococcus* [21], *S. acidocaldarius* [22], *Staphylothermus* [23], and *Thermococcus* [24]. The catalytic center of the enzyme is composed of an incomplete $(\beta/\alpha)7$ folded barrel structure, including $\beta4$-glutamic acid as the catalytic nucleophilic reagent and $\beta7$-aspartic acid as the proton donor, and containing five conserved structural domains [13]. The type II pullulanase has dual activity in hydrolyzing $\alpha$-1,4 and $\alpha$-1,6 glycosidic bonds [25].

Pullulanase required for industrial production usually needs to have good stability and activity at high temperatures for long periods [26]. The improvement of the thermal stability of pullulanase is mainly achieved through truncation of the N/C-terminal structural domain, targeted mutagenesis, and computational design assistance [27,28].

Type I pullulanase has an essential role in the production of glucose and maltose syrups [5]. For example, this enzyme improves the efficiency of starch hydrolysis, reduces the use of saccharification enzymes, shortens reaction times, and improves the purity of the products [29,30]. Because of its ability to act on both $\alpha$-1,6 and $\alpha$-1,4 glycosidic bonds, type II pullulanase is used alone during starch liquefaction and saccharification, thereby preventing the addition of other amylases [31]. Pullulanase added during beer brewing can also reduce costs and enhance product quality. By contrast, the use of pullulanase during resistant starch preparation can improve the properties of cyclodextrins and thus increase their hydrophobicity [32]. Owing to its unique properties, an alkaline-resistant branched-chain amylase is added to detergents, which greatly improves their cleaning efficiency when acting together with alkaline $\alpha$-amylase [33]. Some pullulanases are combined with glycolytic enzymes and used as plaque inhibitors [34]. Thus, pullulanases are currently an essential ingredient in the food industry, healthcare industry, light industry, and green environmental protection industry.

Pul-HJ21 was obtained from *Thermococcus siculi*, and it was isolated and purified at the deep-sea hydrothermal mouth of the Pacific Ocean by Prof. Wang in the laboratory in the early stage. The genus *T. siculi* was first isolated and purified in a seawater-containing lake in Vulcano, Italy [35]. *T. siculi* HJ21 can grow at 60 to 94 °C, with an optimum growth temperature of 88 °C. The pullulanase produced by *Thermococcus siculi* HJ21 has an optimum enzyme activity temperature of 100 °C [24]. In industrial production, a higher action temperature often reduces energy waste and increases productivity. Therefore, we used the Pul-HJ21 gene of *T. siculi* HJ21 to investigate its heat resistance mechanism. In this study, Pul-HJ21 was truncated. The sequence analysis, 3D structure modeling, conserved active site analysis, and related enzymatic property analysis of the truncated pullulanase Pul-HJΔ782 were also carried out. The temperature and pH stability of Pul-

HJΔ782 improved with the truncation of the C-terminus of Pul-HJ21. At the same time, C-terminus truncation reduced the spatial obstruction during substrate binding. This also allowed the truncated Pul-HJΔ782 to exhibit higher enzymatic digestion efficiency of the substrate. This study provides a theoretical basis for the modification of type II pullulanase for stability and the application of this modified pullulanase for industrial purposes.

## 2. Results and Discussion

### 2.1. Pullulanase Bioinformatics Analysis

The physicochemical properties of Pul-HJ21 and Pul-HJΔ782 are listed in Table 1. The ProParam analysis revealed that the truncated pullulanase has a molecular weight of 90,046.50 Da, an extinction coefficient of 210,970 (mol·cm)$^{-1}$, an amino acid isoelectric point of 4.65, and positively and negatively charged residues of 120 and 69, respectively. The stability analysis with an instability coefficient of 25.79 revealed that the protein has excellent stability, and, therefore, the protein has great potential for industrial applications. The Pul-HJΔ782 three-dimensional model constructed using Alpha Fold2 was scored using UCLA-DOE LAB-SAVES v6.0 software. The pull-down plot showed that more than 95% amino acids were located in the allowed region. Based on the scoring results, we determined that the protein model could be used for further experiments such as molecular docking and molecular dynamics simulations.

We compared the structure of Pul-HJΔ782 with the GH57 family glucan branching enzyme from *Thermococcus kodakaraensis* (magenta; PDB code 3N98), and we found that they shared a similar triangular structure [36,37]. The (β/α)7 barrel domain is shown in orange, the double helix domain is shown in purple, and the C-terminal domain is shown in yellow (Figure 1b,c). Two key catalytic residues, Glu318 and Asp421, are contained in the (β/α)7 barrel domain. The structure of the full-length Pul-HJ21, the structure of Pul-HJΔ782, and the incomplete (β/α)7 folded barrel structure are shown in Figure 1. The central (β/α)7 barrel structure is also a common feature of GH57 family amylases [38].

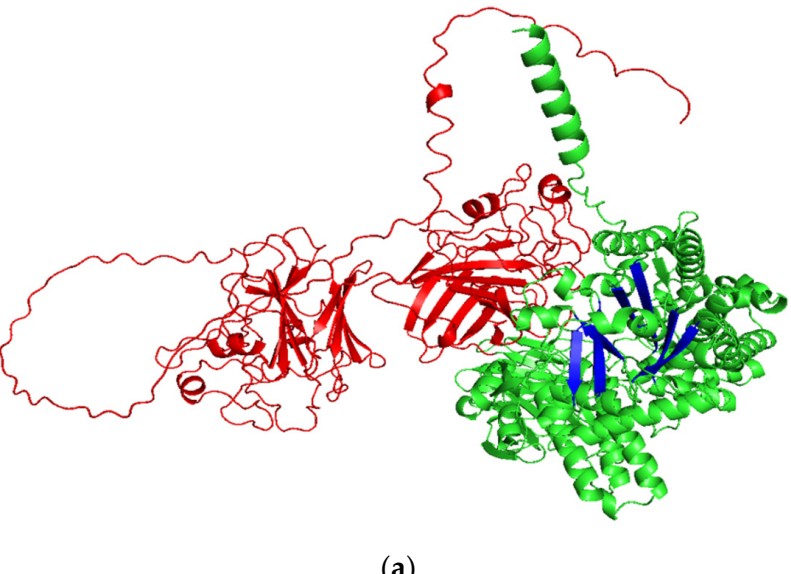

(**a**)

**Figure 1.** *Cont.*

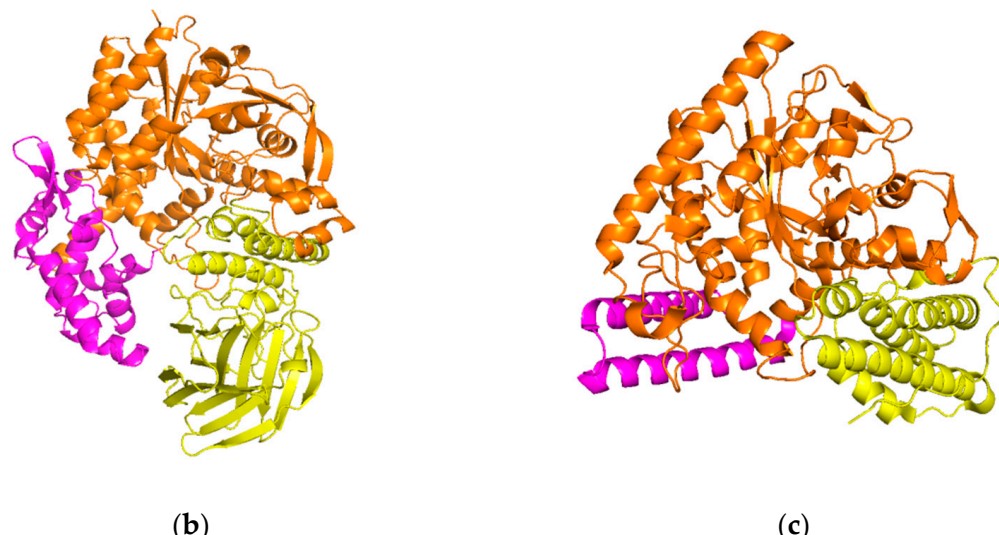

(**b**)                                                              (**c**)

**Figure 1.** (**a**) Structural mode of Pul-HJ21. The green part shows the three-dimensional structure of Pul-HJΔ782; the red part indicates the deleted structure of Pul-HJ21; the blue color indicates the incomplete $(\beta/\alpha)_7$ folded barrel structure of Pul-HJΔ782. (**b**) Structural mode of Pul-HJΔ782. The $(\beta/\alpha)_7$ barrel domain is shown in orange, the double helix domain is shown in purple, and the C-terminal domain is shown in yellow. (**c**) Structural mode of Pul-HJΔ782. The $(\beta/\alpha)_7$ barrel domain is shown in orange, the double helix domain is shown in purple, and the C-terminal domain is shown in yellow.

**Table 1.** Comparison of the physical and chemical properties of truncated enzymes and Pul-HJ21.

| Characteristics | Pul-HJΔ782 | Pul-HJ21 |
|---|---|---|
| Number of amino acids | 782 | 1372 |
| Molecular weight (Da) | 90,046.50 | 153,712.72 |
| Theoretical pI | 4.65 | 4.58 |
| Asp + Glu | 120 | 198 |
| Arg + Lys | 69 | 109 |
| Formula | $C_{411}H_{6159}N_{1021}O_{1218}S_{21}$ | $C_{6972}H_{10528}N_{1758}O_{2107}S_{32}$ |
| Instability index | 25.79 | 28.83 |
| Ext. coefficient/$(\text{mol·cm})^{-1}$ | 210,970 | 311,240 |
| Aliphatic index | 82.26 | 81.78 |
| Grand average of hydropathicity (GRAVY) | −0.410 | −0.37 |

*2.2. Expression of Truncated Pullulanase*

According to the search results of the conserved domain in NCBI and Inter PRO database comparison, the protein consists of 1372 amino acids. The primary function of the C-terminal functional region of the conserved domain structure was predicted as carbohydrate transport and metabolism and signal transduction. The protein was predicted to have six possible structural domains: the GH57 family conserved structural domain of COG1449 and Glyco_hydro_57, the carbohydrate-binding structural domain, the N-terminal catalytic structural domain of GH57N-APU heat-activated branched-chain, the glycoside hydrolase barrel structural domain of Glyco_hydro/deAcase, and the α-amylase-related structural domain. In this protein, the C-terminal structural domain is primarily composed of the carbohydrate-binding functional COG4945 structural domain. We also predicted the CGP-CTERM domain at positions 1331–1350. The structural domain has an essentially invariant motif, Cys-Gly-Pro, followed by a highly hydrophobic transmembrane domain, which is always located at the C-terminus of the protein (Figure 2). In Pang et al.'s study of the catalytic properties of *Pyrococcus yayanosii* CH1, the excised N-terminal domain increased the thermal stability and hydrolysis efficiency of the pullulanase enzyme [18].

Therefore, we designed different primers to truncate Pul-HJ21 according to the different lengths of the structural domains and investigated the changes in enzyme properties after truncation for different conserved domains at the N-terminal end.

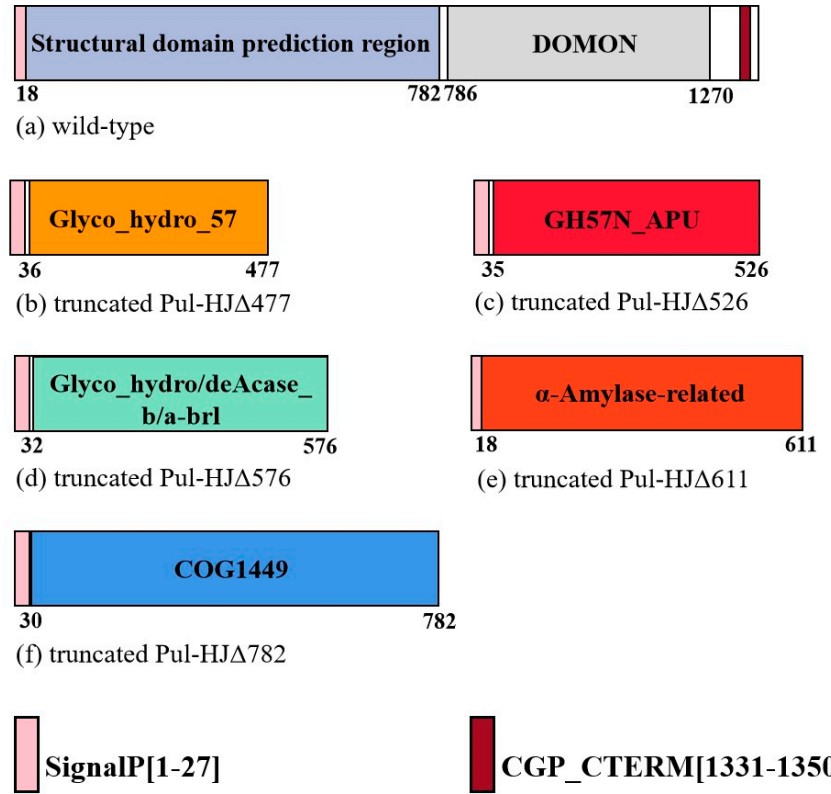

**Figure 2.** Sequence analysis of HJ21 and different truncates.

We ligated five truncates, Pul-HJΔ782, Pul-HJΔ636, Pul-HJΔ576, Pul-HJΔ526, and Pul-HJΔ477, to the vectors pET-29a(+) BamHI and NdeI. The resulting recombinant plasmids were transformed into BL21. Induced expression revealed that only Pul-HJΔ782 was enzymatically active. We speculate the presence of an essential structure at the C-terminus of Pul-HJΔ782 that maintains the pullulanase and α-amylase activities. Moreover, this structure has a vital role in the stability and binding to the substrate. The deletion of this structure would cause the complete loss of enzymatic activity, as in the case of C-terminus truncation by Kim et al. [39].

*2.3. Characteristics of Truncated Pullulanase*

By using pullulan and soluble starch as substrates, we measured the sum of the enzymatic activities of the enzyme from 50 °C to 120 °C in 50 mM Tris–HCl buffer. The results of experiments on temperature stability before and after truncation and the optimum temperature are plotted in Figure 3. When the C-terminus was truncated, the optimum temperature of 100 °C almost remained the same (Figure 3a,b), but the stability at 100 °C improved. Before truncation, the enzyme activity decreased to 62% of the untreated group after 1 h treatment at 100 °C, and only 30% of the enzyme activity remained after 5 h treatment (Figure 3c). After truncation, the loss of enzyme activity after 1 h treatment at 100 °C was only lower than that before truncation. The loss was less than 15% after 1 h of treatment at 100 °C, and the residual enzyme activity of 57% remained after 5 h of treatment (Figure 3d). Thus, the C-terminal sequence was refined to improve the temperature. The half-life of the Pul-HJΔ782 mutant was calculated to increase by 1.32 h at 100 °C. The half-life at 80 °C also increased to 63.01 h from 15.40 h after truncation (Table 2).

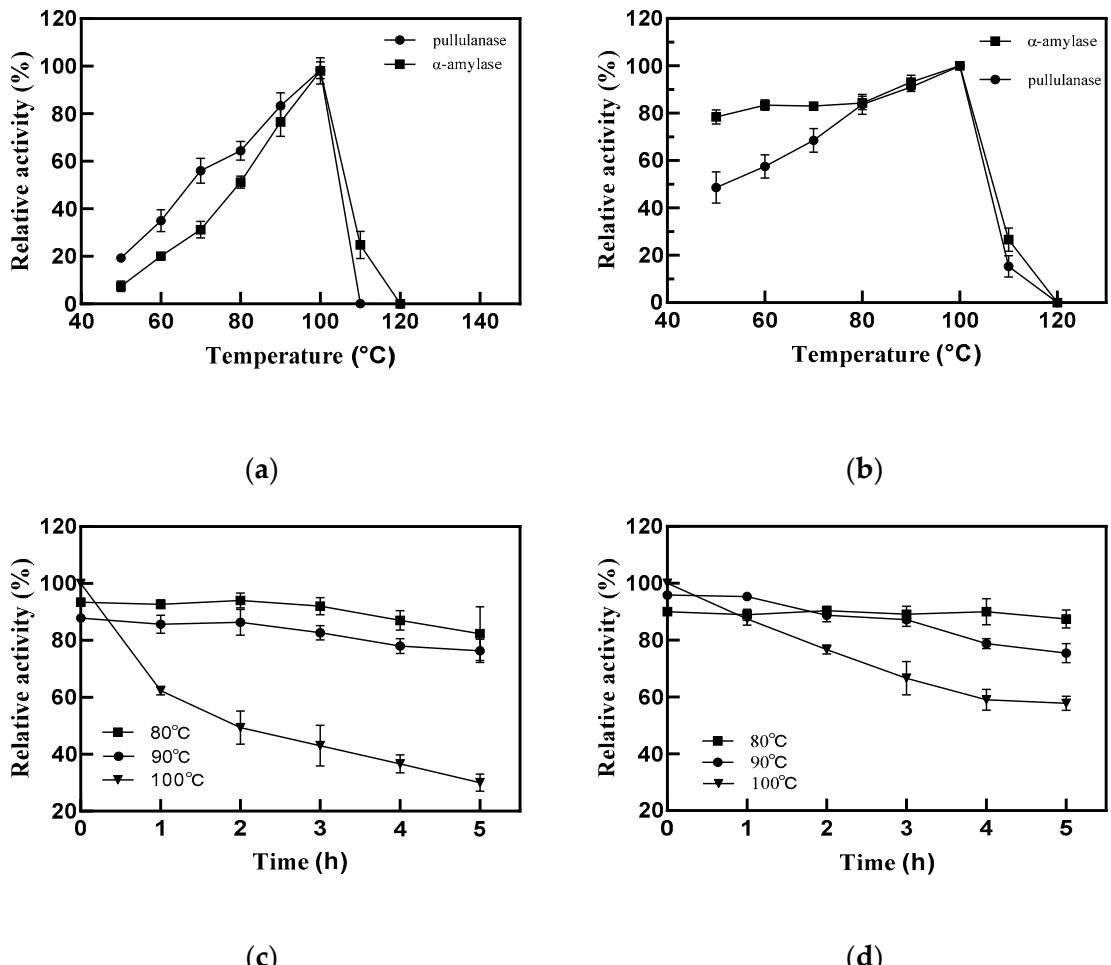

**Figure 3.** Temperature properties of Pul-HJ21 and Pul-HJΔ782. (**a**) Optimum temperature of Pul-HJ21; (**b**) optimum temperature of Pul-HJΔ782; (**c**) temperature stability of Pul-HJ21 based on the sum of α-amylase and pullulanase activities; (**d**) temperature stability of Pul-HJΔ782 based on the sum of α-amylase and pullulanase activities.

**Table 2.** Half-life of Pul-HJ21 and Pul-HJΔ782.

| Mutant and Temperature | $k_d$ [a] | $T_{1/2}$ [b] |
|---|---|---|
| Pul-HJ21-100 °C | 0.176 ± 0.010 | 3.95 ± 0.3 |
| Pul-HJ21-90 °C | 0.051 ± 0.003 | 13.64 ± 1.1 |
| Pul-HJ21-80 °C | 0.019 ± 0.003 | 37.41 ± 8.4 |
| Pul-HJΔ782-100 °C | 0.132 ± 0.001 | 5.25 ± 0.1 |
| Pul-HJΔ782-90 °C | 0.048 ± 0.012 | 15.40 ± 5.4 |
| Pul-HJΔ782-80 °C | 0.011 ± 0.001 | 63.54 ± 8.2 |

Values represent the mean of three independent sets of experiments with SD < 5%. [a]: first-order rate constants of inactivation. [b]: half-life = ln2/kd.

Before truncation, the optimal pH for α-amylase activity was 6 and that for pullulanase activity was 6.5 (Figure 4a). After truncation, the optimal pH for α-amylase activity was 4.5 and that for pullulanase activity was 6.5 (Figure 4b). Both enzymes before and after truncation maintained the best stability at pH 7 (Figure 4c,d). The broad pH tolerance range of Pul-HJΔ782 significantly improved after truncation compared with that before truncation, and the stability of α-amylase in the pH range 4–8 was more than 80% (Figure 4d).

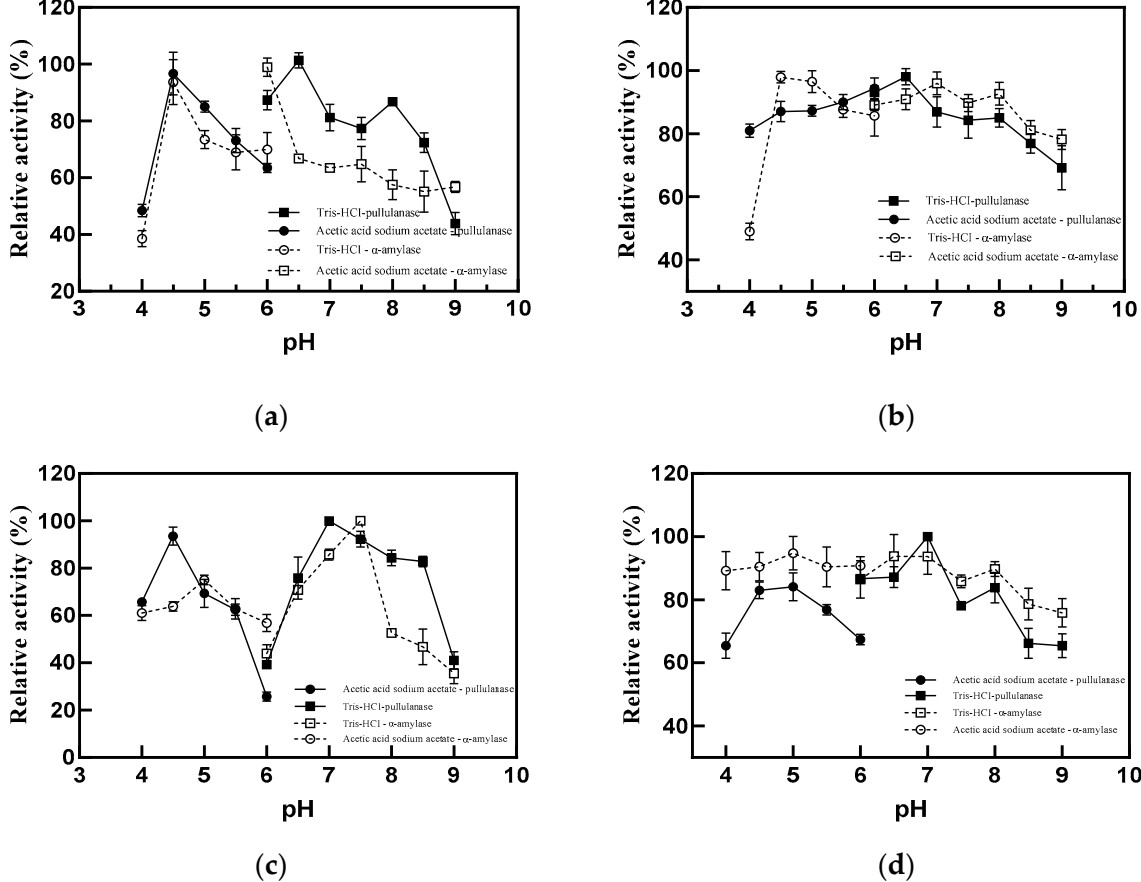

**Figure 4.** pH properties of Pul-HJ21 and Pul-HJΔ782. (**a**) Optimal pH of Pul-HJ21; (**b**) optimal pH of Pul-HJΔ782; (**c**) pH stability of Pul-HJ21; (**d**) pH stability of Pul-HJΔ782.

The high stability and wide pH tolerance of Pul-HJΔ782 at high temperatures allows it to adapt to most starch liquefaction production processes. Moreover, it can shorten the process time and save costs by improving the hydrolysis efficiency. Thus, it has great potential for industrial production.

*2.4. Hydrolysates of Truncated Pullulanase*

The hydrolysates of Pul-HJΔ782 were analyzed to study the effect of truncation on products. We performed HPLC analysis of the hydrolysates of pullulanase and soluble starch degraded by Pul-HJ21 and Pul-HJΔ782 at 90 °C for different times. The results are listed in Tables 3 and 4 and Figure 5. The standard curve of oligosaccharides by HPLC measurements is shown in Table S1. The calibration plots of peak value (height) versus concentration were linear for all standards ($R^2 > 0.99$, $p = 0.01$). When soluble starch was used as the substrate, glucose appeared as the minimal product. The minimal substrate for the reaction was maltotriose when pullulan was used as a substrate. Therefore, the changes in the different minimal products were detected up to 5 h of the reaction. When pullulan was the substrate, the maltotriose content was basically unchanged in 5 h, being 264.86% before truncation and 264.28% after truncation, respectively. When soluble starch was the substrate, the glucose content in the 5 h reaction increased from 44.39% before truncation to 68.23% after truncation. Therefore, the truncated Pul-HJΔ782 had higher hydrolysis efficiency.

**Table 3.** Concentration of hydrolysates after the enzymatic digestion of soluble starch at different times by Pul-HJ21 and Pul-HJΔ782.

| Enzyme and Time | Oligosaccharides (mg/mL) | | | | |
|---|---|---|---|---|---|
| | G7 | G5 | G3 | G2 | G1 |
| Pul-HJ21-15 min | 11.21 ± 0.48 | 0.78 ± 0.05 | 0.11 ± 0.03 | 0.12 ± 0.03 | 0.41 ± 0.02 |
| Pul-HJ21-30 min | 12.06 ± 0.38 | 0.76 ± 0.01 | 0.13 ± 0.02 | 0.19 ± 0.01 | 0.70 ± 0.20 |
| Pul-HJ21-1 h | 12.37 ± 0.30 | 0.92 ± 0.07 | 0.34 ± 0.05 | 0.38 ± 0.03 | 0.46 ± 0.04 |
| Pul-HJ21-3 h | 13.64 ± 0.17 | 1.09 ± 0.07 | 0.58 ± 0.14 | 1.06 ± 0.10 | 0.87 ± 0.02 |
| Pul-HJ21-5 h | 11.94 ± 0.72 | 0.88 ± 0.15 | 0.55 ± 0.17 | 0.85 ± 0.40 | 1.32 ± 0.04 |
| Pul-HJΔ782-15 min | 11.27 ± 0.39 | 0.82 ± 0.03 | 0.18 ± 0.10 | 0.28 ± 0.11 | 0.68 ± 0.05 |
| Pul-HJΔ782-30 min | 10.71 ± 0.61 | 0.76 ± 0.20 | 0.11 ± 0.03 | 0.35 ± 0.03 | 0.63 ± 0.10 |
| Pul-HJΔ782-1 h | 11.84 ± 0.53 | 0.96 ± 0.10 | 0.12 ± 0.01 | 0.28 ± 0.03 | 1.00 ± 0.05 |
| Pul-HJΔ782-3 h | 11.14 ± 0.33 | 0.64 ± 0.40 | 0.19 ± 0.03 | 0.36 ± 0.03 | 1.92 ± 0.39 |
| Pul-HJΔ782-5 h | 13.39 ± 0.56 | 1.17 ± 0.03 | 0.35 ± 0.11 | 0.60 ± 0.05 | 3.00 ± 0.18 |

**Table 4.** Concentration of hydrolysates after the enzymatic digestion of pullulan at different times by Pul-HJ21 and Pul-HJΔ782.

| Enzyme and Time | Oligosaccharides (mg/mL) | | | | |
|---|---|---|---|---|---|
| | G7 | G5 | G3 | G2 | G1 |
| Pul-HJ21-15 min | 13.12 ± 0.56 | 0.95 ± 0.01 | 0.37 ± 0.14 | / | 0.42 ± 0.01 |
| Pul-HJ21-30 min | 14.62 ± 0.09 | 0.98 ± 0.02 | 0.50 ± 0.03 | / | 0.47 ± 0.03 |
| Pul-HJ21-1 h | 16.61 ± 0.72 | 0.88 ± 0.01 | 0.81 ± 0.09 | / | 0.48 ± 0.01 |
| Pul-HJ21-3 h | 14.88 ± 0.30 | 0.87 ± 0.03 | 2.03 ± 0.17 | / | 0.43 ± 0.03 |
| Pul-HJ21-5 h | 19.60 ± 0.78 | 1.06 ± 0.02 | 5.27 ± 0.28 | / | 0.52 ± 0.03 |
| Pul-HJΔ782-15 min | 13.86 ± 0.68 | 0.84 ± 0.02 | 0.14 ± 0.01 | / | 0.14 ± 0.01 |
| Pul-HJΔ782-30 min | 15.54 ± 0.74 | 0.81 ± 0.02 | 0.18 ± 0.03 | / | 0.11 ± 0.03 |
| Pul-HJΔ782-1 h | 17.84 ± 1.95 | 0.32 ± 0.73 | 0.36 ± 0.01 | / | 0.11 ± 0.02 |
| Pul-HJΔ782-3 h | 19.52 ± 0.91 | 0.75 ± 0.03 | 1.22 ± 0.13 | / | 0.14 ± 0.02 |
| Pul-HJΔ782-5 h | 18.90 ± 0.47 | 0.69 ± 0.69 | 1.99 ± 0.02 | / | 0.15 ± 0.02 |

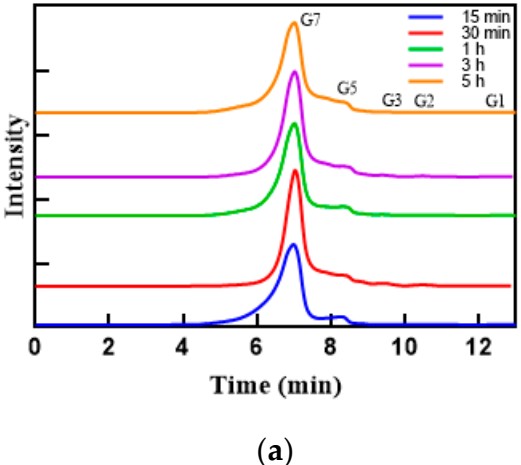

(**a**)

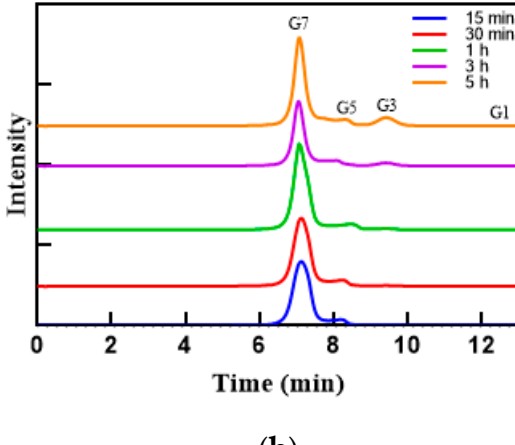

(**b**)

**Figure 5.** *Cont.*

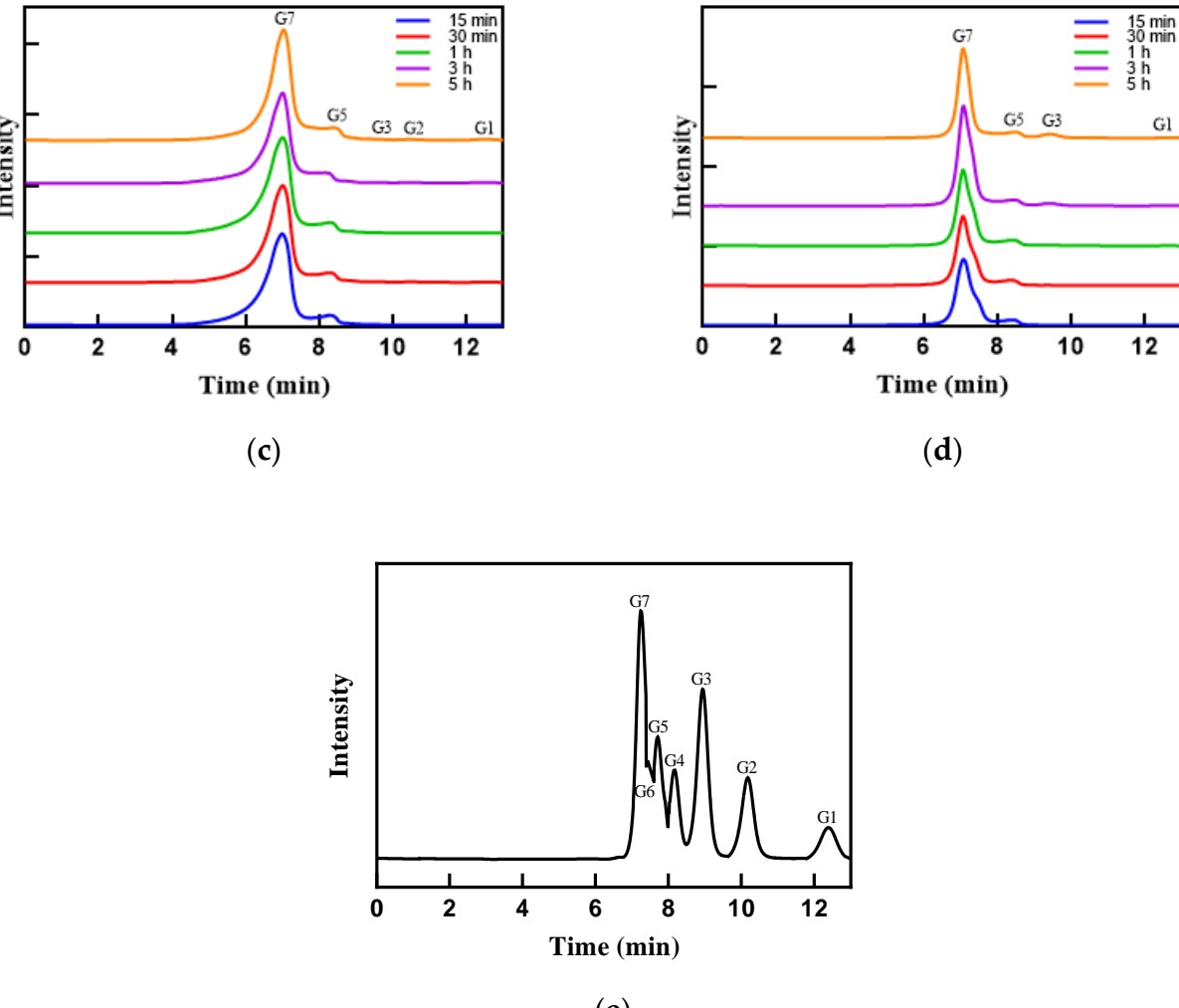

**Figure 5.** HPLC plots of Pul-HJ21 and Pul-HJΔ782 for the enzymatic digestion of pullulanase and soluble starch at different times. (**a**) HPLC analysis of Pul-HJ21 during the enzymatic digestion of soluble starch at different times; (**b**) HPLC analysis of Pul-HJ21 during enzymatic digestion of pullulan at different times; (**c**) HPLC analysis of Pul-HJΔ782 during the enzymatic digestion of soluble starch at different times; (**d**) HPLC analysis of Pul-HJΔ782 during the enzymatic digestion of pullulan at different times; and (**e**) elution peaks of reference standards.

### 2.5. Molecular Mechanism of Truncated Pullulanase

By comparing the B-factor values of Pul-HJ21 and Pul-HJΔ782 (Figure 6A), we found that the flexible loop ring region of Pul-HJΔ782 located at Leu259-Glu265 had decreased significantly. This indicated that the rigidity of the flexible loop at 259L-265E was enhanced after truncation (Figure 6B). Meanwhile, 259L-265E are located within the second conserved region of the GH57 family (Figure 7). Glu265, as one of the most conserved amino acid residues, is also the candidate site for targeted mutagenesis studies [19]. Usually, increasing the rigidity of a protein at its flexible site is also considered an important tool in protein stability modification.

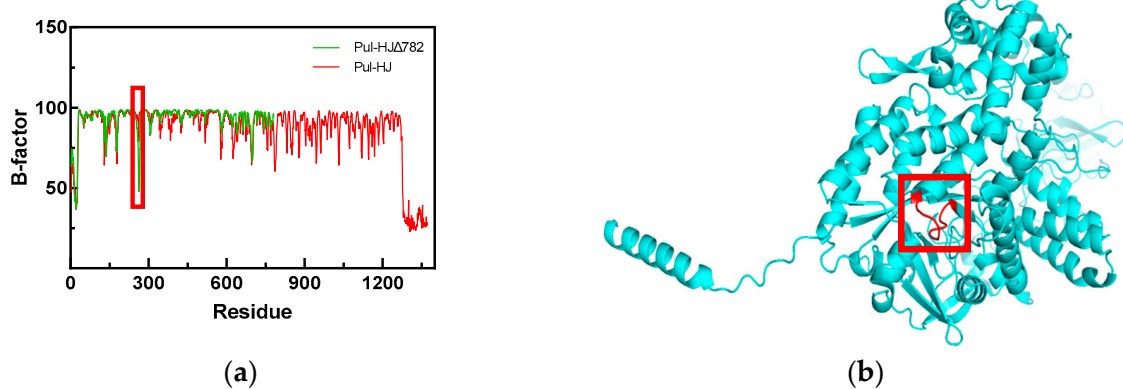

**Figure 6.** (**a**) B-factor values of Pul-HJ21 and Pul-HJΔ782; (**b**) the loop region with reduced B-value after truncation.

```
Pul-HJ△782/1-782      1 MRRVV - ALLLAVLMVGSL IGANVKTVGAAEPKPLNV I IVWHQHQPYYYDP IQDIY   54
AAD28552.1/1-1337     1 MRRVV - ALF I A ILMLGS IVGANVKSVGAAEPKPLNV I IVWHQHQPYYYDPVQDVY   54
BAC10983.1/1-1089     1 MKKGL - AMFL IFLVALS IAEVG- - -VKAEEPKPLNV I IVWHQHQPYYYDP IQDIY   51
AAL82059.1/1-985      1 MSRKL - SLLLVFL IFGSMLGAN - NIVKAEEPKPLNV I IVWHQHQPYYYDPVQG IY   53
CAB49104.1/1-1362     1 MRGKFRAL FVLV IFV ISVLGPG IKSVNAAEPKPLNV I IVWHQHQPYYYDP ILGVY   55
                                                                      CSR-1

Pul-HJ△782/1-782    219 KALYDKVDEGGYTRDDVKTVLDAQLWLLNHTFEEHEKVNLLLGNGNVEVTVVPYA  273
AAD28552.1/1-1337   219 KALYDKVDEGGYTRADVKTVLDAQ IWLLNHTFEEHEK INLLLGNGNVEVTVVPYA  273
BAC10983.1/1-1089   216 KALYDKVDVGGYTKEDVATVLKHQMWLLNHTFEEHEK INYLLGNGNVEVTVVPYA  270
AAL82059.1/1-985    218 KALYDKVDEGGYTREDLKTVLYHQMWLLNNTFKEHEK INLLLGNGNVEVTVVPYA  272
CAB49104.1/1-1362   220 KALYDKVDTGGYTRKDVETVLKHQMWLLNHTFEEHEK INLLLGNGNVEVTVVPYT  274
                                                                #      CSR-2

Pul-HJ△782/1-782    274 HP IGP ILNDFGWEGDFNDQVKRADELYKQYLGNGTAVPVGGWAAESALNDKTLE I  328
AAD28552.1/1-1337   274 HP IGP ILNDFGWDSDFNDQVKKADELYKPYLGGGTAVPKGGWAAESALNDKTLE I  328
BAC10983.1/1-1089   271 HP IGPLLNDFGWYEDFDAHVKKAHELYKKYLGDNRVEPQGGWAAESALNDKTLE I  325
AAL82059.1/1-985    273 HP IGP ILNDFGWSEDFDAHVKKAHELYKKYLGGGVATPRGGWAAESALNDKTLE I  327
CAB49104.1/1-1362   275 HP IGP ILNDFGWYEDFDAQVKKANELYKEYLGAGKVTPKGGWAAESALNDKTLE I  329
                                                                #      CSR-3

Pul-HJ△782/1-782    384 FTYAGMNQYQAVDDFVNELLK IQKENYDGSLVYVVTLDGENPWEHYPYDGK IFLT  438
AAD28552.1/1-1337   384 FTYSGMNQQQAVEDFVNELLKLQKQNYDGSLVYVVTLDGENPVENYPYDGELFLT  438
BAC10983.1/1-1089   381 FRYSGMNQYQAVEDFVNELLKVQKENYDGSLVYVVTLDGENPWEHYPFDGK IFLE  435
AAL82059.1/1-985    383 FRYSGMNQYEAVEDF INELLK IQKYNYDGSLVYV ITLDGENPWEHYPYDGKLFLE  437
CAB49104.1/1-1362   385 FRYAGMNQYDAVKNFVEELLK IQKQNYDGSLVYV ITLDGENPWEHYPFDGKLFLE  439
                                                                       CSR-4

Pul-HJ△782/1-782    547 KKDEMSQEDWEKAHEYLLRAEASDWFWWYGSDQNSGQDFTFDRYLKTYLYEMYRL  601
AAD28552.1/1-1337   547 NKDKMSQADWEKAYEYLLRAEASDWFWWYGSDQDSGQDYTFDRYLKTYLYEMYKL  601
BAC10983.1/1-1089   545 NKNKV - -VDWNTAYEYLLRAEASDWFWWYGSDQDSGQDYTFDRYLKTYLYEMYKF  597
AAL82059.1/1-985    547 NKDKMDSASWEKAYEYLLRAEASDWFWWYGNDQDSGQDYSFDRYFKTYLYE IYKL  601
CAB49104.1/1-1362   549 NKDNV - -KDWNKAYEYLFRAEGSDWFWWYGRDQNSMQDYVFDRYKLYLYE IYKL  601
                                          CSR-5
```

**Figure 7.** Conserved sequence regions (CSR) in the family GH-57. # indicates the catalytically active site.

Pul-HJΔ782 was sequenced against the conserved sequences of different enzymes from the GH57 family (GenPept: AAD28552.1; BAC10983.1; AAL82059.1; CAB49104.1) by sequence alignment [19]. Based on the description of R. Zona et al., we identified five conserved sequence regions (CSR) from the GH57 family (Figure 7). CSR-1 is the His42-Gln43-Pro44 co-identifier sequence. The Glu266 of CSR-2 is also defined as one of the most conserved amino acids. CSR-2 and CSR-3 contain two catalytic residues, and Glu318 is identified as the catalytic nucleophilic residue and Asp421 as the proton donor [40]. A higher conserved Asp566 is present in CSR-5, which was also shown to interact with two active-site water molecules [19]. It is noteworthy that in the structural model, CSR-2 was located in a flexible ring, and the rigidity of this flexible ring was enhanced after truncation. Consequently, the stability and catalytic efficiency of Pul-HJΔ782 were changed. In their study of the glycogen branching enzyme from GH57, S. Na et al. found that the flexible ring next to the active site had an important role in the overall catalytic reaction and the flexible

ring was directly involved in the catalytic reaction [38]. For 259L-265E in Pul-HJΔ782, they did not construct a catalytic pocket with Glu318 and Asp421, but 259L-265E connects the α-helix and β-fold of the (β/α)7 barrel structure. The increased rigidity in the 259L-265E range also suggested that the (β/α)7 barrel structure will have better backbone support. Combined with our results, we found that a more stable catalytic backbone could lead to higher stability and better catalytic efficiency in catalytic reactions at higher temperatures.

Using AutoDock, we then performed molecular docking with Pul-HJΔ782 as the ligand isomaltotriose as the acceptor. The docking results showing the active region are presented in Figure 8. All ligand conformations generated were scored on the basis of the degree of binding (kcal/mol). The best position chosen, −9.2 kcal/mol, was analyzed for the forces formed between the ligand and receptors. In the results of AutoDock, Glu318, Ser569, Tyr46, Asp570, Ser581, Gly131, Arg56, Asp135, Trp58, His62, Gln43, Asn424, and Asp421 were involved in hydrogen bonding as proton donors and acceptors. Among the ligands, Trp573, Trp574, Gln583, Pro130, Pro44, His42, and His274 formed hydrophobic interactions with the acceptor (Figure 9). Of them, Glu318 and Asp421 coincided with the catalytic site of type II pullulanase already reported [19,41–43]. Moreover, in the GH57 enzyme-specific position, His40 was identified as involved in the donor 1 subsite, and His42 was indirectly involved in catalytic reactions via water molecules [40,41]. Combined with the result that the conserved catalytic site appears upon docking with isomaltotriose, the smallest unit produced during the enzymatic cleavage of pullulanase, we speculate that Pul-HJΔ782 also has pullulanase hydrolase activity [2].

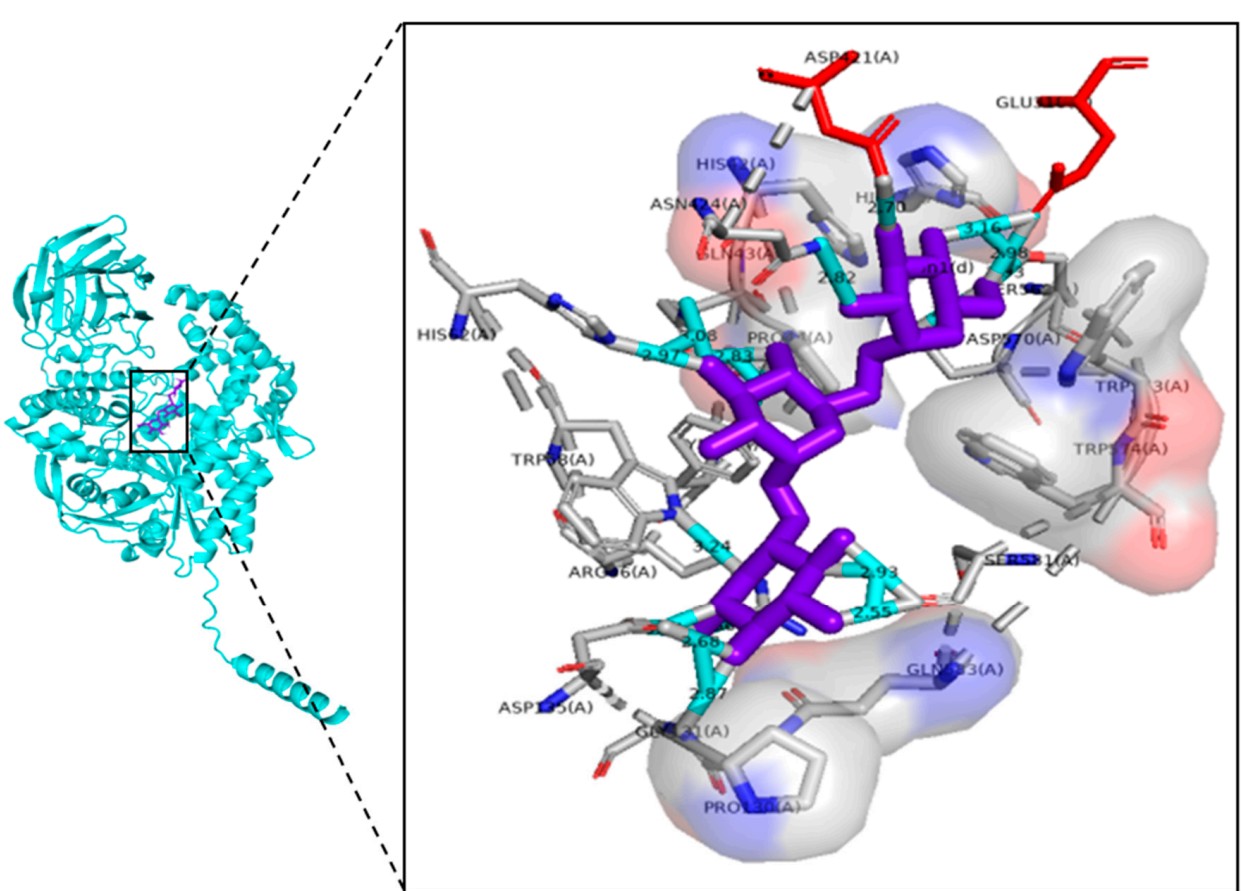

**Figure 8.** Pul-HJΔ782 molecular docking results. The purple-colored structure is isomaltotriose.

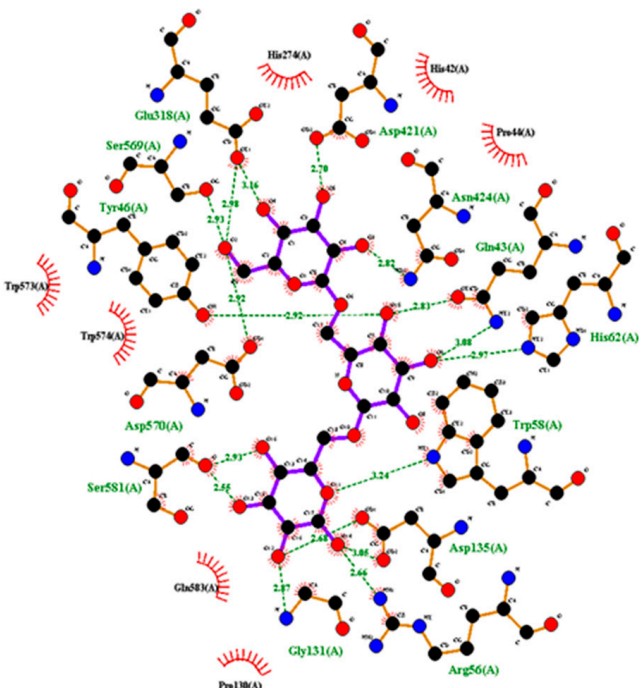

**Figure 9.** Force formed between ligands and receptors and amino acid sites when Pul-HJΔ782 uses isomaltotriose as a substrate.

We truncated the C-terminus of Pul-HJ21 and the protein isoelectric point was changed. Usually, changes in the protein isoelectric point are considered one of the main causes of pH changes. With truncation, the catalytic efficiency of Pul-HJΔ782 for the substrate, especially pullulanase, substantially increased. A larger spatial structure near the active site was possibly exposed after the C-terminal structural domain of Pul-HJ21 was deleted. Because a larger space may allow the more rapid recognition of more substrates, the reaction rate increased accordingly. This finding also provides a theoretical basis for the subsequent modification of other pullulanases.

## 3. Materials and Methods

### 3.1. Materials and Reagents

#### 3.1.1. Strain and Plasmid

*Thermococcus siculi* HJ21 was obtained from our laboratory. *Escherichia coli* BL21(DE3) and *E. coli* DH5α receptor cells were purchased from Solaibao. pET29a was purchased from Biotech Bioengineering Co., Ltd. (Shanghai, China).

#### 3.1.2. Reagents

A DNA marker (Biotech Bioengineering Co., Ltd., Shanghai, China), Gel Red (Takara Biomedical Technology Co., Ltd., Beijing, China), agarose (Biowest Biotechnology Co., Ltd., Beijing, China), pullulan (Biowest Biotechnology Co., Ltd., Beijing, China), FastPure Plasmid Mini Kit and FastPure Gel DNA Extraction Mini Kit, and ClonExpress MultiS One Step Cloning Kit (Nanjing Novozymes Biotechnology Co., Ltd., Nanjing, China) were purchased for the experiments. The other reagents used in the experiments were analytically pure and were from Sinopharm Chemical Reagent Co., Ltd. (Shanghai, China). The primers used in the experiment are listed in Table 5 and were synthesized by Biotech Bioengineering Co., Ltd. (Shanghai, China).

**Table 5.** Primers used in the experiment.

| Primer | (5′ to 3′) | Primer |
|--------|-----------|--------|
| His-F | atgcaccaccaccaccaccacAGGCGGGTGGTTGCC | His-F |
| T-His-F | taagaaggagatatacatatgATGCACCACCACCACCACCACAGG | T-His-F |
| 477-R | acggagctcgaattcggatccCATCATCTTGGGGGTGAGCTT | 477-R |
| 526-R | acggagctcgaattcggatccGCCTATCCAGGTGGAGAGCG | 526-R |
| 576-R | acggagctcgaattcggatccCCCATACCACCAGAACCAGTCG | 576-R |
| 636-R | acggagctcgaattcggatccATTCTTCATCTCCCCCTCCTTG | 636-R |
| 782-R | acggagctcgaattcggatccCTTCAGCTCGACGGGGGT | 782-R |

*3.2. Methods*

3.2.1. Pullulanase Gene Acquisition

The sequence of the pullulanase gene in *T. siculi* HJ21 was checked in GenBank under the registration number EU849120 and downloaded. The sequence was cloned using SnapGene 5.2 [44] to simulate the design of adding two enzyme cleavage sites, BamHI and NdeI. The finalized gene sequence was synthesized by Biotech Bioengineering Co., Ltd., China. The synthesized pullulanase gene was ligated to the pET-29a(+) plasmid.

3.2.2. Bioinformatics Analysis of Pullulanase

Pullulanase gene sequences were spliced and compared using SnapGene 5.2, and conserved domain and ORF analyses of protein families were performed using the NCBI database and Inter Pro (https://www.ncbi.nlm.nih.gov/ (accessed on 16 February 2023), http://www.ebi.ac.uk/interpro/ (accessed on 16 February 2023), respectively). The secondary structure of pullulanase was predicted through NetSurfP analysis (https://services.healthtech.dtu.dk/service.php?NetSurfP-2.0 (accessed on 16 February 2023)), and the three-dimensional structure was modeled in three dimensions by using Alpha Fold2 (https://colab.research.google.com/github/deepmind/alphafold/blob/main/notebooks/AlphaFold.ipynb (accessed on 16 February 2023)) [45]. The modeled proteins were submitted to the UCLA-DOE LAB-SAVES v6.0 (https://saves.mbi.ucla.edu/ (accessed on 16 February 2023)) server for scoring and evaluation and then uploaded to SWISS-MODEL (https://swissmodel.expasy.org/interactive (accessed on 16 February 2023)) for the Ramachandran map. The docking of small-molecule glycans and protein receptors was performed using AutoDock-GPU [46], and the docking results were analyzed through Pymol mapping. The physicochemical properties of the proteins were analyzed using ProParam (https://web.expasy.org/protparam/ (accessed on 16 February 2023)). The B-factor value of each amino acid atom in the protein files was modeled in Alpha Fold. Then, we used the ba2r tool to convert the B-factor values of these atoms to the B-factor values of individual amino acids [47].

3.2.3. Truncated Expression of the Pullulanase Gene

Based on the results of the conserved domain analysis, the primers were designed to obtain gene sequences of different lengths. The sequences were ligated to the pET29$\alpha$(+) vector linearized by BamH I and Nde I enzymatic cleavages by using a non-ligase-dependent single-fragment fast cloning kit. The recombinant plasmids containing different target genes of varying sizes were separately transferred into DH5$\alpha$ receptor cells. Then, the successfully transformed positive clones were extracted for plasmid sequencing, whereas the normally sequenced plasmids were re-transferred into BL21 receptor cells. The valid positive clones were inoculated in 10% liquid LB medium containing 50 μg/μL kanamycin and incubated at 37 °C, 180 rpm until the OD$_{600}$ reached 1. Isopropyl β-D-Thiogalactopyranoside (IPTG) was then added at a concentration of 50 μg/μL and fermented at 16 °C for 12 h at 180 rpm. The broth was centrifuged at 8000× *g* for 15 min to collect the pellets. The pellets were then resuspended in PBS buffer and sonicated at 600 W for 15 min by using an ultrasonic disruptor. The disruptor was set for 1 s with a 2 s interval. Finally, the supernatant was collected by centrifugation at 13,200× *g* for 10 min at 4 °C and was considered the crude enzyme solution and stored at 4 °C.

### 3.2.4. Detection of Pullulanase Activity

One enzyme activity unit (U) is defined as the amount of enzyme required to produce 1 μmol of maltose through the enzymatic digestion of pullulan in 1 min. Enzyme activity was determined using the dinitrosalicylic acid (DNS) method. In this method, 50 μL of crude enzyme solution was added to a test tube containing 150 μL of 1% pullulanase or 1% soluble starch, and the reaction was carried out at 100 °C for 15 min. Then, to terminate the reaction, 200 μL of DNS solution was added and the mixture was boiled in boiling water for 5 min. Subsequently, 3 mL of pure water was added, and the absorbance value was measured at 540 nm after mixing. The enzyme activity was calculated according to the following formula. When the temperature was equal to or greater than 100 °C, the buffer and enzyme solution were encapsulated in an oil bath in an ampoule. The ampoule was quickly cooled in ice water before adding DNS to terminate the reaction when the oil bath reaction was finished. To prevent the evaporation of water at high temperatures, 1 mL of enzyme solution was encapsulated in an ampoule for the temperature stability assay.

$$\text{Pullulanase enzyme activity}(\text{U/mL}) = \frac{\frac{\Delta OD - b}{a} \times 1000 \times 1000}{342.3 \times 15 \times 50} \tag{1}$$

a: slope in the DNS standard curve; b: intercept in the DNS standard curve.

### 3.2.5. Analysis of Enzymatic Properties

To determine the optimum temperature for pullulanase, its enzyme activity was measured at 50, 60, 70, 80, 90, 100, 110, and 120 °C. In the optimum pH assay, the enzyme activity of pullulanase at pH 4.0–9.0 was determined using 50 mM acetic acid–sodium acetate (pH 4.0–6.0) and Tris hydrochloride (pH 6.0–9.0) as buffers.

In the heat stability assay, the enzyme solutions were incubated at 80, 90, and 100 °C for 5 h, with samples taken every 1 h and then stored at 4 °C. The untreated group was used as the control to determine the enzyme activity. The thermostability was calculated based on the sum of the amylase and pullulanase activities. In the pH stability assay, the enzyme solutions were placed in different pH buffers for 4 h. The untreated group was used as the control to determine the enzyme activity at different pH and to determine the effect of pH on the enzyme solutions.

To determine the half-life, 1 mL of enzyme solution was placed at 80 and 100 °C for different times. After treatment, the enzyme solution was centrifuged at $13,200\times g$ for 30 min to remove the heat-denatured proteins. The residual enzyme activity was then measured. The time was plotted against ln(enzyme activity) and a linear fit was performed. The slope in the linear equation that we then obtained was the primary inactivation constant of the enzyme at that temperature. The half-life was calculated from the following equation.

$$t_{1/2} = \frac{\ln 2}{k_d} \tag{2}$$

### 3.2.6. Analysis of Enzymatic Hydrolysates

The enzyme solution was mixed with the substrate at a 1:2 volume in an equal volume of Tris–HCl buffer (pH 6.0) and incubated at 90 °C for 15 min, 30 min, 1 h, 3 h, and 5 h. All samples were filtered through a 0.22-μm filter membrane (Thermo Fisher Scientific Co., Ltd. (Shanghai, China), and the hydrolysates were detected at 75 °C using a Water Sugar-Pak1 column, where pure water was used as the mobile phase at a flow rate of 0.4 mL/min. For determination of the standard curve of oligosaccharides, different concentrations of seven standard sugar stock solutions were measured by HPLC. Then, the linear relationship was determined between the peak height and concentration.

### 3.2.7. Data Analysis

All experiments were performed in triplicate. All statistical analyses were performed using OriginPro 2021 (64-bit). The error bars presented on the figures correspond to the standard deviations.

## 4. Conclusions

We here developed a C-terminal truncation strategy by analyzing the Pul-HJ21 structural domain. Pul-HJΔ782, Pul-HJΔ636, Pul-HJΔ576, Pul-HJΔ526, and Pul-HJΔ477 were expressed. Only Pul-HJΔ782 was enzymatically active among the five mutants. Pul-HJΔ782 is a type II pullulanase. The thermal stability and pH tolerance of Pul-HJΔ782 improved after truncation. The enzyme retention ability increased from 62% to 87% after 1 h of treatment at the optimum temperature of 100 °C and from 30% to 57% after 5 h of treatment. The α-amylase activity of Pul-HJΔ782 achieved more than 80% stability at pH 4–8, thereby indicating that the truncated sequence at the C-terminus of the structural domain can improve the stability significantly. Pul-HJΔ782 can break down pullulanase into isomaltotriose faster than Pul-HJ21. Thus, the experiments provide an essential reference for the stability of pullulanase and the improvement of its product decomposition efficiency. Moreover, it offers theoretical data to support the development and research of more heat-resistant pullulanases for industrial applications.

**Supplementary Materials:** The following supporting information can be downloaded at: https://www.mdpi.com/article/10.3390/catal13030453/s1, Table S1: Linearity of the HPLC measurements; Table S2: Protein sequences.

**Author Contributions:** X.W.: Conceptualization, methodology, writing—original draft preparation. R.S. and B.D.: Data curation, resources. B.W. and M.L. (Mingwang Liu): Visualization, investigation and software, validation. X.W. and M.L. (Mingsheng Lyu): Writing—review and editing. S.W. and J.L.: Funding acquisition, project administration. All authors have read and agreed to the published version of the manuscript.

**Funding:** This study was supported by the National Key R&D Program of China (Grant No. 2022YFC2805101); the National Natural Science Foundation of China (Grant No. 32172154 and 32201964); 521 Program Grant No. LYG06521202107; the Priority Academic Program Development of Jiangsu Higher Education Institutions (PAPD); and the Research & Practice Innovation Program of Jiangsu (SJCX22_1659).

**Data Availability Statement:** The interaction data used to support the study findings are included within the article and the supporting information file. Moreover, all the data used to support the study findings are available from the corresponding author upon request.

**Conflicts of Interest:** The authors confirm that there is no conflict of interest regarding this paper.

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
