# Peer review of "Improved Stability and Hydrolysates of Hyperthermophilic GH57 Type II Pullulanase from the Deep-Sea Archaeon Thermococcus siculi HJ21 by Truncation"

_catalysts, doi:10.3390/catal13030453_

Round 1

Reviewer 1 Report (Previous Reviewer 2)

It is great to see that the authors resubmit this manuscript even though it was rejected previously. As compare with the previous version, the current manuscript has improved a lot in terms of the data showing and English writing. Yet, there are still some improvements can be made, especially for the current way of illustration for Figure 2, which is still unsatisfactory to me and remains confusing to readers. Please refer below for the detail suggestion for figure 2 and other minor comments.

Major comment:
1) Figure 2, the current way of drawing is still very confusing to readers, please split the figures into (a) wild-type; (b) truncated Pul-HJΔ782; (c) truncated Pul-HJΔ636; (d) truncated Pul-HJΔ526; (e) truncated Pul-HJΔ477. For the wild-type, you can mark a region X and specify region X can be either of the 5 possible structural domain predicted by InterPro. For each truncation, you can draw the respective structural domain in the figure, which can clearly show that the truncation was made according to the predicted structural domain. Lastly, the figure length must be in a relative ratio of the amino acid sequence length and put a ruler in the figure. Please refer to the previous reviewer's comment for the figure example.

Minor comments:

2) L139, 'maintains the pullulanase and α-amylase activities.

3) L143-166, Figure 3 and 4 have subfigure of (a), (b), (c), and (d), please specify clearly in the text which subfigure to refer to, instead of just mentioning Figure 3 and Figure 4.

4) L154, remove 'and pH tolerance' as this paragraph is about optimal temperature and thermostability.

5) L169, (c): Temperature stability of Pul-HJ21 based on the sum of α-amylase and pullulanase activities; (d): Temperature stability of Pul-HJΔ782 based on the sum of α-amylase and pullulanase activities

6) Table 2: include the half-life for 90°C

7) L186 and L188, changed to 'the maltotriose content' and 'the glucose content', remove the terms 'growth rate'

Author Response

Reviewer 2 Report (New Reviewer)

General comments:

This manuscript, in my best opinion, has a good potential to become a solid contribution to the field of amylolytic enzymes with a focus on amylopullulanases and, in particular, on the alpha-amylase family GH57. Unfortunately, the way as the paper has been finally acomplished and presented, drastically reduces its eventual value. It, in fact, suffers from the fact the Authors have ignored (or overlooked, or both...?) most of what has already been achieved in this field. Authors have to place their enzyme, which is the member of the alpha-amylase family GH57, into the overall context of the family GH57, taking into account and discussing appropriately with relevant literature sources concerning the tertiary structures, catalytic machinery, sequence analyses, evolutionary studies, etc., etc. All this has simply been ignored or the paper has been written as if nothing from that has been known. This is unacceptable because such a presentation would simply introduce a confusion into the scientific lietrature. On the other hand, the experimental science itself seems to be all right and even well-executed, which is a positive news and can serve as a fundament for the highly-valued future paper.

Major and specific points:

(1) Title. The information should be added that this enzyme belongs to the family GH57, e.g., "...hyperthermophilic GH57 type II pullulanase from..."

(2) Abstract. The reader has to be informed also in the abstract that the main subject of this study, the Pul-HJ21, is a member of the alpha-amylase family GH57; i.e. to be clear already in the abstract that it is not a GH13 pullulanase/amylopullulanase.

(3) Introduction, L32. "Amylase" is a vague term; Authors have to be clear in their expressions.

(4) Introduction, L37-39. The first sentence is obviously erroneous: "pullulanase is classified into ... amylases."...?! The next sentence contains also clear inconsistency: "Different pullulanases ... exhibit different catalytic mechanisms..." - this might be true only if one specifies the families GH13 and GH57 because within a family, enzymes possess the same catalytic machinery. Moreover, both families GH57 and GH13 employs the same retaining catalytic mechanism.

(5) Introduction, L47-56. The entire paragraph has to be thoroughly rewritten. It is not simple how to criticize that text since it consists of a mixture of correct information and misleading incorrect details. Authors have to clearly distinguish pullulanasess type II (amylopullulanases) originated from family either GH13 or GH57, the three-dimensional structure of their catalytic domain fold (TIM-barrel = beta/alpha-8-barrel in GH13 vs the incomplete TIM-barrel = beta/alpha-7-barrel in GH57), different catalytic machineries (Asp/beta4, Glu/beta5 and Asp/beta7 in GH13 vs Glu/beta4 and Asp/beta5 in GH57), different number and sequences of identified conserved sequence regions (CSRs), presence of starch-binding domains (SBDs) known as CBM families (mainly CBM48 and CBM41 for pullulanases from GH13 - subfamilies GH13_12, GH13_14 and GH13_14), etc., etc. There is a wealth of literature sources and also even reviews summarising the families GH13 and GH57; the same for SBDs (easily identifiable in literature databases). Of course, Authors are not asked to write a detailed minireview, but, on the other hand, some relevant introduction that would clearly name the things and, especially, places the PulHJ-21 into the concept of the CAZy families, i.e. the family GH57, is really necessary.

(6) Introduction, L71-76. Again, Authors do not distinguish between pullulanases and amylopullulanases as well as between members of the family GH13 (subfamilies GH13_12, 13 and 14) and those of the family GH57. That should clearly be done so.

(7) Results. It is strange, but the manuscript does not contain, in fact, any discussion. There are only 1. Introduction, 2. Results, 3. Materials and Methods and 4. Conclusion. So the part "2. Results" has to be completed by a relevant discussion, but not only in the title of the heading "2. Results and Discussion", but also thorough the text of the entire part of the paper. In addition to that, the text has not only to be rewritten and improved, but also supported by adequate References, which the manuscript in its current version lacks a lot, such as - focused on the family GH57 - tertiary structure studies, in silico sequence/specificity and evolutionary studies, structure/function studies; papers delivering experimental charaxcterization of individual amylopullulanases in the family GH57 - but not only being listed in the References, but rather being discussed with the results achieved in the present study.

(8) Figure 1. It shows, in fact, nothing (the left part - the structure itself). It has to be deleted and replaced by, e.g., a ribbon model emphasizing at least its incomplete TIM-barrel with succeeding bundle of alpha-helices (i.e. the catalytic part of the protein); but if the AlphaFold was able to build the model of the entire truncated protein, then also the remaining parts of the Pul-HJdelta782 should be shown.

(9) Figure 2. Authors should pay more attention in order to prepare a more convincing domain arrangemwent for their enzyme. Again, there is wealth of information about these enzymes so it cannot be accepted to extract all the data only from automatic servers and databases annotating proteins sequences. Authors should take into account the literature delivering many details about the family GH57 members and extract the data for their own study.

(10) Results, L120. What is the term "the carbnohydrate-binding structural domain" based on...? What exactly does it mean...? A CBM...? If yes, a relevant reference should support that statement. There is just one example of CBM20 in GH57 putative amylopullulanase from Kosmotoga olearia pointed out in the literature; if that is the case, a reference is missing.

(11) Results, L121-122. In the family GH57, there is no "beta/alpha-8-barrel structural domain" adopted for the catalytic domain fold; instead, there is the beta/alpha-7-barrel catalytic domain (i.e. the incomplete TIM-barrel; please certify also the L228 where "TIM-barrel" is mentioned) in the family GH57! This point clearly demonstrates how, unfortunately, Authors treated their manuscript laxly.

(12) Results, L127-130. This sentence refer to a citation incorrectly; it is not Bi et al., but Pang et al. [27].

(13) Results, L130-132. This sentence is a bit out of the context since it refers to a study dealing with a family GH13 amylopullulanase (the subfamily GH13_14) [28].

(14) Results, L204. The fact that the Pul-HJ21 (and its truncated versions) are type II pullulanases has been known also from published in silico analyses of the family GH57 since, currently, there are cca 10 experimentally characterized GH57 pullulanases type II that are in this family known as amylopullulanases; Authors should check again the literature and dealt with the sources satisfactorily.

(15) Figure 6b. If the loop region emphasized in the picture is unique for this enzyme (or if it is characteristic of all GH57 pullulanases type II), such an observation should be accompanied also by a relevant amino acid sequence alignment.

(16) Figure 7. What has been the rationale for selection of the sequences aligned in the picture...? It has to be explained. Moroever, the alignemtn has to be completed by highlighting the 5 CSRs well-accepted in the family GH57 taking into account and discussing the unique sequence features of amylopullulanases (important unique sequence positions, i.e. residues, the so-called sequence fingerprints). Based on what were the residues, additional to the two forming the catalytic machinery (Glu and Asp), mentioned here...?

(17) Figure 8. The detail on the right has to be somehow re-done since as it is now, one caanot see any details that, however, are very important to be visible.

(18) Methods, L270. The GenBank accession No. for Pul-HJ21 should be written without a space: "EU849120"; maybe Authors could consider to use also the GenPept Protein ID#, which is more unambiguous and has also been used in the CAZy database: "ACJ03924.1".

Round 2

Reviewer 2 Report (New Reviewer)

General comments:

In my opinion, the manuscript has been much improved and the overall presentation of the story has been significantly clarified. My recommendation is, however, still to perform a minor revision that means mainly to complete the list of additional important/appropriate/relevant References.

Minor specific points:

(1) The reference to theCAZy database - currently the Ref. No. [8] Lombard et al. 2014 should be replaced by the most actual one: Drula et al. 2022 - DOI: 10.1093/nar/gkab1045.

(2) The Ref. No. [14] Nisha & Satyanarayana 2013 - although being of interest and deserved to be referred to - is not most appropriate to support most of statements concerning the family GH57 and their sequence-structural details as well as the presence of CBM41/CBM48/CBM68 in various pullulanases from the family GH13. For example, there have been published several detailed analyses focused directly on the family GH57 and similarly several insightful reviews on CBM families of amylolytic enzymes, respectively. Authors should - as suggested already in the first round of the peer-review - consult with those studies and referred to accordingly in order the informed reader can follow the individual/particular message of the present study easily.

(3) L47-70: Typical family GH13 pullulanases have been classified into three subfamilies - GH13_12, GH13_13 and GH13_14; Authors have mentioned only two - GH13_12 and GH13_13; this has to be completed; suggested reading from literature search, e.g., DOI: 10.1002/prot.25309, and eventually other papers.

(4) L130: "GH57 family branched enzyme" - it is the "GH57 family glucan branching enzyme" (not "branched").

Author Response

This manuscript is a resubmission of an earlier submission. The following is a list of the peer review reports and author responses from that submission.

Round 1

Reviewer 1 Report

The manuscript by Wu et al. describes the construction and characterization of truncated versions of a Type II pullulanase from Thermociccus siculi. In this work, the authors mention the construction of truncations of the gene at different positions after the N terminal pullulanase catalytic domain. However, they report that only one of these constructions was active, which almost exactly coincides with the expressed fragment in a previous work from one of the coauthors of the manuscript (Jiao, et al. Current Microbiology (2011) 62, 222–228). Although in this new version the authors included this reference, they just pointed it out as the work in which the enzyme was isolated from its source, and they do not discuss the constructions and the characterization made, which were just a few residues away from the ones reported here (residue 791 in that work, vs residue 782 in the present work). In the cover letter, they mention that the results of this new truncated version have higher stability than the previously truncated protein, but they look the same. Both keep about 60% activity after 5 h incubation at 100°C

I find serious problems with the experimental part of the manuscript and the data interpretation, besides a negligent data presentation, which makes me reject the manuscript for publication in Catalyst.

The authors only made some esthetic corrections, but the points that were relevant and that may imply repeating some experiments were not attended.

The list of comments would be quite long, but I will address some that I consider the most relevant to make my point.

First, why the previously truncated protein, reported in the article by Jiao, et al. is ignored, and what is the justification to construct a slightly shorter version of that protein? And be honest, and report that this turned out to behave the same.

In materials and methods, it is described the determination of activity as reducing sugars expressed as equivalents of maltose, however, I did not find any report of the activity of the truncated and wild-type enzymes. This data is relevant to make a comparison between the two enzymes. In this same point, the formula used to determine reducing sugars uses the glucose molecular weight, instead of that from maltose.

A point not addressed from the previous version: Although in Section 3.2.5 a correct description of the pH range for buffering of acetic/acetate and Tris base/TrisHCl pairs is done, labels in figures 4c and 4d do not correspond to the correct buffering range. Acetic acid can buffer between pH 3.7-5.7, and Tris between pH 6.5-8.5. Besides, the change of buffer seems to affect the activity, which was completely ignored by the authors. The ups and downs of activity upon change of buffer system impede drawing any conclusion about the pH profile of these enzymes. The effect was less marked for the truncated version, but still, the lack of coincidence in the activities at the same pH when changing the buffer, suggests that the buffer affects the activity. These experiments should be repeated using a buffer system in which both buffers are present throughout the pH range studied.

The product profile shown in the chromatograms in figure 5, together with tables 3 and 4 leaves more questions than answers. First, the values in the table not only changed as far as the amounts reported previously, but also some oligosacharides that were reported previously are not present in this new version, and vice versa. 

The resolution of the chromatograms prevents an accurate estimation of oligosaccharides from G5 to G7. The mobile phase should be adjusted to improve resolution in this zone or even consider running a gradient.

The authors show a chromatogram with standard oligonucleotides from G1 to G7. Since they have a standard of each of these oligosaccharides, they should evaluate the amount of each in the samples generated at different reaction times, instead of showing the results as % (Which, by the way, is never described how it was calculated. I assume that is % of the total area in the chromatogram). The results as presented are meaningless since they are not accompanied by how many reducing sugars were produced in each case, to get an idea of the progress of the reaction. The data analysis as presented contributes to seeing changes in the % of oligosaccharides without following a clear trend. I think if the experiments were carefully conducted, the estimation of the amount of each oligosaccharide can be determined and it would give more information about how the hydrolysis is progressing, the reaction profile of each enzyme, and if there is some transglycosylation activity, and when this starts being relevant.

Continuing with this piece of data, the data in Tables 2 and 3 are not congruent with the chromatograms shown, at least at 5 h of reaction.

Line 255 “With truncation, the catalytic efficiency of Pul-HJΔ782 for the substrate, especially pullulanase, substantially increased.”. There is no data to support this conclusion. Quantification of reducing sugars should accompany the HPLC data, and instead of using % area, the amount of each sugar should be evaluated to make a fair comparison between the two enzymes

Other minor points:

In line 40 the authors say “Maltose and  maltotriose are used by pullulanase type I as minimal products for the breakdown of starch or other polysaccharides.” It is not clear if the authors mean that maltotriose and maltose are the minimal products obtained by pollulanase action, or if they are the minimal substrates used by the enzyme.

In line 90 the authors state “Therefore, we used the Pul-HJ21 gene pair of Thermococcus siculi HJ21 in an attempt to investigate its heat resistance mechanism”. What does it mean the Pul-H121 gene pair was used? Are there two genes?

The caption in Figure 1 should state clearly that it is a structural model and not a 3D structure.

The caption in Figure 2 requires more explanation. What is on the left, and what is on the right side? Are these different enzymes?

 In lines 339-240, the authors describe that “The thermostability was calculated based on the sum of the amylase and pullulanase activities.” How come the thermostability can be estimated by the sum of two individual activities?

The manuscript urgently needs an edition, since it is plagued with grammar errors.

Reviewer 2 Report

Specific comments:

  1. Introduction: The information is sufficient but unorganized. The authors should restructure the whole section, remove irrelevant information, and focus mainly on Type II pullulanase.

  2. Table 1, remove other truncated enzymes information since they had no enzyme activity.

  3. The authors had amended figure 2. Yet, the current version is still unclear. Please refer below example. Consider removing other truncations and other non-relevant predicted domains.

  1. L124, “This protein contains six conserved domains: …”. Please amend the elaboration accordingly as the current information is misleading.

  2. The authors truncated the enzymes based on the five predicted structural domains, obviously only one predicted domain is functional. Wouldn’t it better to show only the protein structure of the successful clone, in which this solves the comments 3 and 4 above at the same time.

Reviewer 3 Report

Thank you for addressing all the comment from the first revision. Authors are still advise to revise the language before final submission.